# Response of Sea Water Exchange Processes to Monsoons in Jiaozhou Bay, China

Zhenhuan Tian [1,2,3], Jinghao Shi [1], Yuanyuan Liu [4], Wei Wang [2,3,*], Chunhua Liu [5], Fangfang Li [6] and Yanqin Shao [2,3]

1   College of Marine Geosciences, Ocean University of China, Qingdao 266100, China; tt266003@aliyun.com (Z.T.); estuary2@ouc.edu.cn (J.S.)
2   No. 1 Institute of Geology and Mineral Resources of Shandong Province, Jinan 250100, China; binghen1205@gmail.com
3   Shandong Engineering Laboratory for High-Grade Iron Ore Exploration and Exploitation, Jinan 250100, China
4   Shandong Polytechnic College, Jining 272067, China; gakkiw@aliyun.com
5   Shandong Institute of Geological Survey, Jinan 250013, China; chunhua_liu321@aliyun.com
6   Aerospace Information Research Institute, Chinese Academy of Sciences, Beijing 100190, China; ffli1@mail.ie.ac.cn
*   Correspondence: www86_0@aliyun.com; Tel.: +86-531-8859-6392

**Abstract:** The self-purification capacity of semi-closed bays is closely related to the exchange process of open sea water. In recent years, with the enhancement of human development activities, environmental problems such as eutrophication, weak hydrodynamics, and poor water exchange capacity have appeared in the bays. In this paper, the water exchange time and flow field in Jiaozhou Bay (JZB) were investigated using the environmental fluid dynamics code with a coupled dye module. Specifically, Jiaozhou Bay was divided into seven zones to explore the effect mechanism of a monsoon on the water exchange process. A detailed analysis was performed on the current water exchange status in the highly polluted northeastern region of the bay and its influence on the surrounding areas. Based on the definition of the average residence time and considering the effect of the tracer release moment, the distribution of the water exchange time in the bay under three circumstances was obtained. Results showed that the timing of the tracer release exerted minimal influence on the average residence time. The water exchange process was influenced by a combination of astronomical and meteorological factors. The overall exchange capacity of the bay was strongest under the impact of a winter monsoon and tides, followed by a summer monsoon and tides, and the weakest exchange occurred under the influence of tides alone. Moreover, both summer and winter monsoons greatly facilitated water exchange in the heavily polluted northeastern region. However, pollutants from this region had a significant impact on surrounding areas during a summer monsoon. Changes in the structure and intensity of residual flow fields were the primary causes of exchange rate discrepancies.

**Keywords:** Jiaozhou Bay; Eulerian dispersion model; water exchange; average residence time; exchange rate discrepancies

## 1. Introduction

Bays and their surrounding areas possess abundant resources and unique environments but these environments are fragile due to intense human activities [1–5]. With ongoing exploitation, environmental issues have emerged in bays to varying degrees, such as weakened hydrodynamics, deteriorated water exchange capabilities, and eutrophication [5–10]. Understanding and addressing these issues are integral parts of research regarding the self-purification capability of bays [11,12]. The self-purification capability of a bay is primarily controlled by biogeochemical and physical processes within a water body. The former primarily reduces the concentration of active pollutants through biological and chemical reactions while the latter involves convective transport, dilution, and diffusion

in a bay. This is a process where the water mixes and exchanges with surrounding and external water bodies, ultimately mitigating the impact of pollutants. The time required for this exchange with the open sea serves as a metric of the water exchange capability of a bay [13,14].

The exchange between an estuary/bay and offshore waters is a complex dynamical problem that influences biogeochemical cycles and ecosystem processes. Many studies show that water exchange between the tide-dominated bay and offshore waters varies seasonally and interannually and is highly influenced by freshwater discharge [15,16] and winds [17]. Among these driving forcings, the dynamics of wind-driven bay-offshore waters exchange are the most widely reported, e.g., in Sansha Bay [18,19], Bohai Bay [20,21], San Francisco Bay [16,22,23], and Chesapeake Bay [24]. Most previous research on tidal and wind-induced water transport was based on numerical model experiments and short-term field observation of hydrodynamic factors. Due to the lack of necessary long-term on-site monitoring, the bay water exchange process with temporal and spatial scale changes under unsteady conditions has not been well understood.

Jiaozhou Bay, located in the southern portion of the Shandong Peninsula in China (Figure 1), is a semi-enclosed bay bordering the Yellow Sea along the line connecting the southern end of Tuandao to the northern end of Jiaozishi. It can be further divided into the inner Jiaozhou Bay and outer Jiaozhou Bay by the east–west line connecting the eastern end of Huangdao to Tuandao [2]. North of the line is Inner Bay and south of the line is Outer Bay. While Jiaozhou Bay is the "Mother Bay" of Qingdao, it has experienced a decline in its marine area, a shrinkage of its tidal volume, and a decrease in its self-purification capability with the progression of urbanization. This has caused gradual environmental degradation, especially in the northeastern water bodies [25,26]. Extensive studies were conducted on the water exchange process in the bay aimed at promoting sustainable development. Based on an analysis of salinity data and using a box model, Wu et al. [27] concluded that the tidal exchange rate between the inner and outer Jiaozhou Bay was 7%, with a half-exchange time of approximately 5 d. Zhao et al. [28] developed a particle tracking model and found that differences in the spatial structure of the tidal residual current field led to a range of average water residence times from as short as 7 d to as long as several months. Liu et al. [29], using an Eulerian dispersion model of conservative substance that considered convective and diffusion processes, found that the average residence time in the bay was approximately 52 d. While the aforementioned studies have deepened our understanding of the water exchange process in Jiaozhou Bay, they also have certain limitations. Some models did not incorporate a comprehensive range of physical processes [27,28] and some did not consider the drying and wetting processes of tidal flats [29]. In addition, most of the existing studies paid more attention to astronomical factors (tidal currents) while giving less consideration to the impact of non-astronomical factors (monsoons) on the water exchange process [30–33].

Over the past few decades, Jiaozhou Bay has witnessed dramatic transformations in its coastline morphology and topography, resulting in subsequent shifts in its hydrodynamic field and water exchange processes [33–35]. This study focuses on the influence of physical processes on the bay's self-purification capability. It leverages more recent data on coastline and water depths and employs a hydrodynamic model that has been well-verified within Jiaozhou Bay. The model encompasses the wetting and drying processes of tidal flats and utilizes a dissolved conservative substance as a tracer for the bay's water. A simulation of the bay's water exchange under the influence of monsoons was then conducted to scrutinize the impact of wind stress and tidal forces on water exchange. This study further discusses the water exchange in severely polluted areas and the environmental implications.

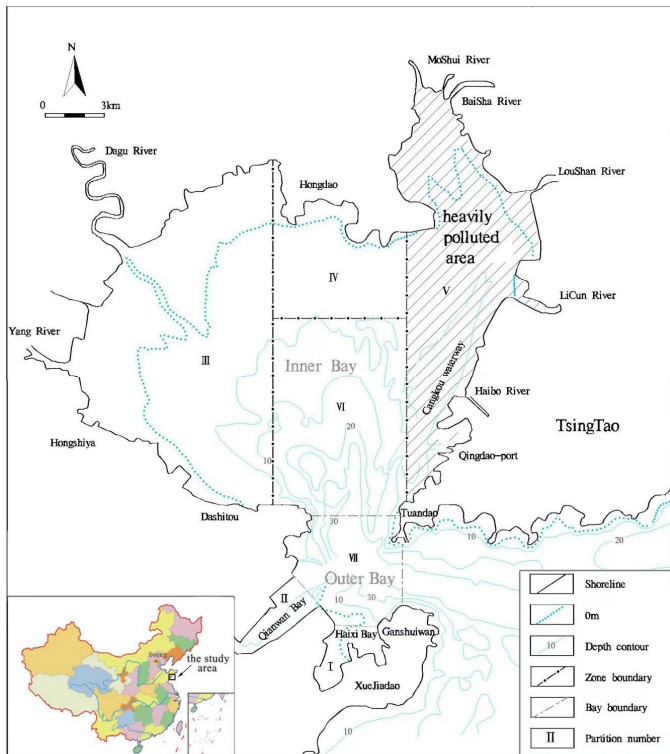

**Figure 1.** Topography and regional division map of Jiaozhou Bay, China.

## 2. Materials and Methods

### 2.1. Modeling

The environmental fluid dynamics code (EFDC) is capable of simulating multi-dimensional flow fields, substance transport, and ecological processes in aquatic environments and its applicability has been validated in numerous marine areas [30,36–42]. The model employs a sigma coordinate transformation in the vertical direction and Cartesian or orthogonal curvilinear coordinates in the horizontal direction. The equations of motion are solved using a combination of finite volume and finite difference methods, with staggered grid discretization in the horizontal direction. Temporal integration is accomplished with a second-order accurate finite difference method along with a technique to separate the barotropic and baroclinic modes. The barotropic mode (external mode) utilizes a semi-implicit calculation method that allows for larger time steps. The baroclinic mode (internal mode) employs an implicit format that considers vertical diffusion. The model allows for drying and wetting in shallow areas using a mass conservation scheme, making it suitable for application in intertidal areas. Detailed information on the principles of the model and its solution methods can be found in Hamrick's article [36].

Dissolved conservative substances serve as excellent tracers for water exchange. The tracer (dye) transport module in the EFDC, which is an Eulerian dispersion model developed based on convection and diffusion equations, is solved using a fractional step scheme involving implicit vertical diffusion and explicit advection and horizontal diffusion, making it particularly suitable for studying bay water exchange problems. Ji et al. [40], Shen et al. [41], Shi et al. [30], Zhang et al. [42], and Xiong et al. [16] successfully utilized this model to investigate water exchange issues in estuaries and bays.

### 2.2. Water Exchange Time

Multiple definitions of the water exchange time exist, leading to a certain level of confusion regarding its terminology and meaning [43]. For instance, the term "residence time" has more than one interpretation. Some researchers define it as the time needed for conservative particles to move from a specific location to the boundary of the computational

domain [44,45] while others define it as the time taken for all the water in a system to be completely exchanged [41,46]. In this study, we adopted the average residence time (ART) proposed by Takeoka to represent the water exchange capability of the bay, which is particularly suitable for studying the self-purification capacity of semi-closed bays where tidal currents serve as the primary driver of water exchange [47–50].

The average residence time ($T_r$) refers to the duration needed for the majority of the conservative tracer substance initially released in the entire or part of a bay to be exchanged into the seawater of the open sea. Let $t_0$ and t denote the initial moment and a specific moment in the exchange process, respectively, and let $C(t)$ denote the concentration of a conservative tracer at a certain moment. For this tracer, the corresponding $T_r$ is given by

$$T_r = \int_0^\infty r(t)dt, \text{ where } r(t) = C(t)/C(t_0),\tag{1}$$

where $r(t)$ is a residual function representing the process of a decrease in the tracer concentration in a water body. If the tracer concentration in a water body decays exponentially, i.e., $C(t) = C(t_0)e^{-\beta t}$, substituting this into Equation (1) results in $T_r = 1/\beta$. $T_r$ is the time required for the tracer concentration to decay to $1/e$ (37%) of the initial concentration. The magnitude of the average residence time, $T_r$, can be used to characterize the water exchange capacity of an entire bay or a local area. The shorter the water exchange time, the stronger the self-purification capacity.

*2.3. Model Construction*

The model used remote sensing images of the shoreline of Jiaozhou Bay acquired in 2012 along with water depth data measured in 2011. The computational domain was a fan-shaped maritime area with its shoreline stretching from the southwest end of Xuejiadao to Shazikou (Figure 2). The open boundary was set on the arc of the southeast open sea, ensuring a substantial buffer zone beyond the mouth of the bay to mitigate the impact of the open boundary. An orthogonal curvilinear meshing method was adopted for better shoreline fitting and for appropriately enhancing the resolution within the bay and the tidal flat areas. The model included a total of 11,623 grid cells, with an average spatial resolution of 279 m.

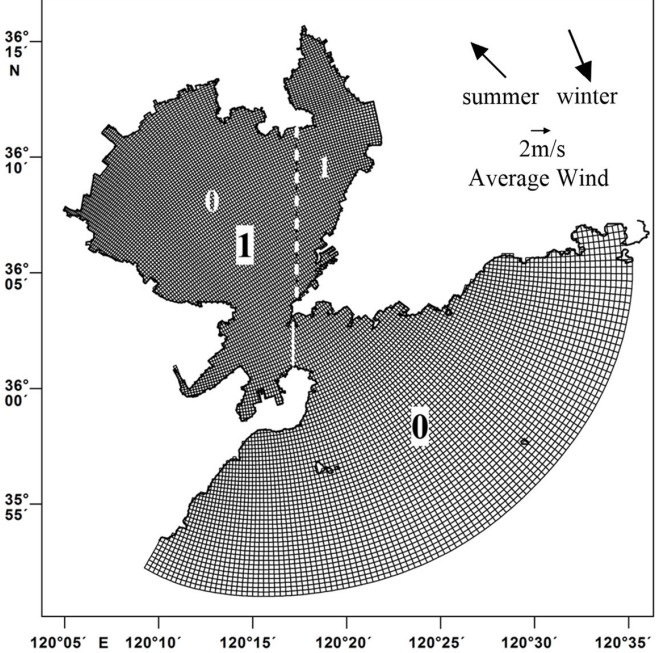

**Figure 2.** Horizontal curvilinear grid of Jiaozhou Bay, China.

Jiaozhou Bay is characterized as a strong tidal bay, with most areas being relatively shallow. The bay mouth and channels, among other deep-water areas, exhibit robust dynamics and there is no significant seasonal stratification in the entire bay, with the baroclinic effect being negligible [28,30,32]. Therefore, the model only considered barotropic processes. The rivers flowing into the bay are mostly mountain streams with rainwater sources and have long dry periods. Coupled with upstream reservoir construction, the run-off volume is extremely limited. Most are now perennially dry, with estuaries also serving as sewage outlets [2]. Affected by the east Asian monsoon, the precipitation in Jiaozhou Bay is mainly concentrated in summer and the average annual precipitation is 687.5 mm. However, in the late 1970s, it changed from a relatively rainy period to a relatively dry period [51]. Moreover, the construction of sponge cities around the bay and the relatively small bay area determine that precipitation is a secondary factor for seawater exchange in Jiaozhou Bay. The small salinity difference inside and outside the bay and the small seasonal difference of water salinity in the bay also prove this point [2]. Hence, the dilution effect of freshwater was not considered in the present study. The initial concentration field of the tracer (dye) in the model was set according to the following principles. The initial concentration of the part of the water body of concern was set to one unit. The concentration in the rest of the water body was set to zero. When the tracer reached the open boundary, its concentration was immediately set to zero. As shown in Figure 2, the initial dye concentration was set to one within the bay mouth area and zero outside.

Jiaozhou Bay experiences a regular semidiurnal tide [2]. The model's open boundary considers five primary tidal constituents (M2, S2, K1, O1, and N2), with harmonic constants referring to existing macro-model results and interpolated with tide gauge data. The study area is primarily influenced by the east Asian monsoon climate. According to long-term observational data [2], prevailing winds in Jiaozhou Bay are from the southeast in spring and summer (Figure 2), with an average wind speed of 5.17 m/s, while winds are from the north-northwest in autumn and winter, with an average wind speed of 5.9 m/s. These constant wind fields were applied in the model, representing average conditions for summer and winter winds.

### 2.4. Model Operation and Validation

The model was successfully applied to the study of tides and currents in Jiaozhou Bay [30,52]. Moreover, by leveraging the tracer transport module we conducted an in-depth exploration of the impact of human activities on the hydrodynamics and water exchange of Jiaozhou Bay. The validity of the dye simulation results was confirmed using the salinity recovery process following heavy rainfall. For a detailed account, please refer to the related article [30]. The results demonstrated that the model is highly effective in reflecting the water exchange process in Jiaozhou Bay.

In the model, the time step was uniformly set to 5 s. The boundary tide level started from zero and gradually increased. After 10 d, the tidal current field stabilized and the influence of the initial conditions was essentially eliminated. Next, the tracer concentration was initialized and the tracer module was included in the computation that lasted for a total of 200 d.

### 3. Results and Discussion

To investigate the impact of monsoons on the water exchange process in Jiaozhou Bay, three scenarios were simulated. The first scenario considered only tidal forces, the second scenario incorporated the combined effect of summer monsoon and tidal forces, and the third scenario included a combination of winter monsoon and tidal forces.

The initial concentration of the tracer set during spring–neap cycles and different tidal phases significantly impacts the water exchange time in the bay [53–55]. Hence, this study sought to account for the variations caused by the timing of tracer release. Consequently, four characteristic moments of tidal movement (flood tide, high-slack tide,

ebb tide, and low-slack tide) were chosen as the initial moments of tracer concentration. The tracer was released separately at the four moments in both spring tides and neap tides for each scenario, totaling 24 simulations across three scenarios (i.e., 4 × 2 × 3). After the 24 simulations, a statistical analysis was performed on the model estimates of the water exchange times.

To facilitate a discussion of differences in the water exchange capability among different parts of Jiaozhou Bay's water body, the bay was divided using dotted lines into seven regions based on the hydrodynamic structure and marine functional zoning: Haixi Bay (I), Qianwan Bay (II) (which borders Huangdao), the northwestern region of inner Jiaozhou Bay (III), the region south of Hongdao (IV), the northeastern region of inner Jiaozhou Bay (V), the central region of inner Jiaozhou Bay (VI), and the central region of outer Jiaozhou Bay (VII), as illustrated in Figure 1. The average residence times and standard deviations for the entire bay and each region under the three scenarios are detailed in Table 1 and Figure 3.

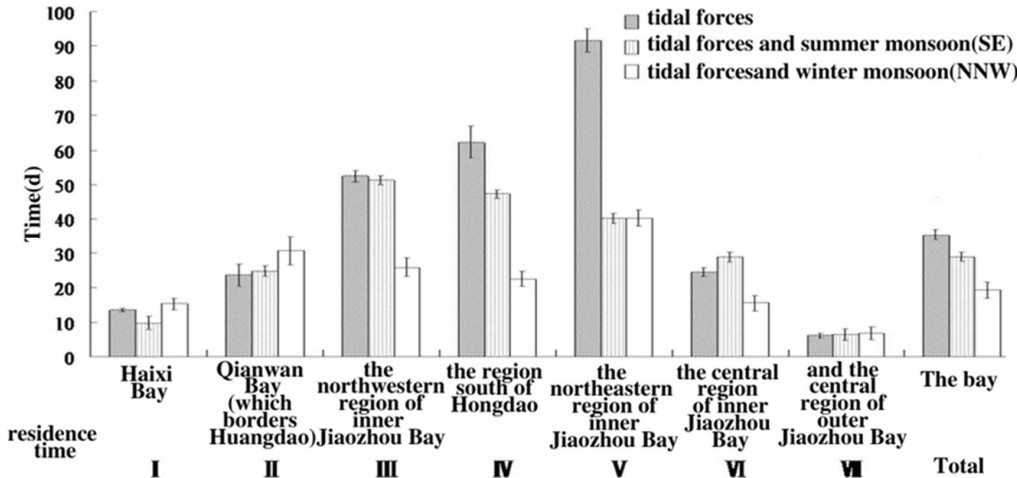

**Figure 3.** Comparison of the average residence time of each partition in Jiaozhou Bay; error bars represent a 95% confidence interval.

The standard deviation of $T_r$ in Table 1 is basically between 0–3 d, indicating that the release of the tracer dye at different tidal phases has a minor effect on $T_r$. However, a comparison of the deviations under different tidal cycles and driving forces revealed that (1) the standard deviation of $T_r$ is greater during the spring tide period than during the neap tide period. It is also greater under the winter monsoon than under the tides alone or the summer monsoon alone; (2) variations in $T_r$ across different regions are substantial. Generally, the stronger the dynamics, the more drastic the changes, and the more significant the influence of the tracer release time. This also reflects the variations in the bay's dynamic field.

The simulation results showed that during flood tides, Yellow Sea water bodies enter Jiaozhou Bay through the inlet, mixing with the water bodies within the bay. During ebb tides, the mixed water mass brought in by the tidal inflow is expelled from the bay, some of which is lost through exchange with the external Yellow Sea water bodies while the rest of the mixed water mass and new Yellow Sea water bodies reenter Jiaozhou Bay with the next flood tide. Thus, the bay's water bodies are gradually replaced by Yellow Sea water bodies. A comparison among the three different conditions (Figure 4) showed that the tracer concentration decreases in a nearly exponential manner over time, albeit at different rates. The decay process displays a semi-monthly oscillation, coinciding with the spring–neap cycles. Water exchange is faster during spring tides and slower during neap tides. This suggests that the presence of monsoons does not fundamentally alter the water exchange process in Jiaozhou Bay and that the tide remains the dominant factor.

**Table 1.** Average values of the residence time of Jiaozhou Bay and 95% confidence intervals are presented in the table.

| | | Haixi Bay | Qianwan Bay (Which Borders Huangdao) | The Northwestern Region of Inner Jiaozhou Bay | The Region South of Hongdao | The Northeastern Region of Inner Jiaozhou Bay | The Central Region of Inner Jiaozhou Bay | And the Central Region of Outer Jiaozhou Bay | The Bay |
|---|---|---|---|---|---|---|---|---|---|
| | **Residence Time** | **I** | **II** | **III** | **IV** | **V** | **VI** | **VII** | **Total** |
| tidal forces | Neap tide | 14.5 ± 0.0 | 25.4 ± 0.4 | 53.9 ± 1.1 | 64.1 ± 1.8 | 92.3 ± 1.5 | 25.7 ± 0.8 | 6.9 ± 0.5 | 36.5 ± 0.9 |
| | Spring tide | 12.6 ± 0.4 | 21.9 ± 2.8 | 51.2 ± 0.6 | 60.5 ± 2.7 | 91.2 ± 1.9 | 23.2 ± 0.4 | 5.4 ± 0.2 | 34.4 ± 0.6 |
| | average | 13.5 ± 0.5 | 23.6 ± 3.2 | 52.6 ± 1.7 | 62.3 ± 4.6 | 91.8 ± 3.4 | 24.5 ± 1.2 | 6.1 ± 0.7 | 35.5 ± 1.5 |
| Tidal forces and summer monsoon | Neap tide | 10.5 ± 0.6 | 25.8 ± 0.1 | 52.3 ± 0.3 | 48.4 ± 0.1 | 41.1 ± 0.2 | 29.9 ± 0.0 | 7.2 ± 0.5 | 29.9 ± 0.1 |
| | Spring tide | 9.0 ± 1.3 | 23.6 ± 1.5 | 50.2 ± 0.9 | 46.2 ± 1.1 | 39.1 ± 1.3 | 27.7 ± 1.3 | 5.7 ± 1.1 | 27.9 ± 1.2 |
| | average | 9.7 ± 1.9 | 24.7 ± 1.6 | 51.2 ± 1.3 | 47.3 ± 1.2 | 40.2 ± 1.5 | 28.8 ± 1.4 | 6.5 ± 1.6 | 28.9 ± 1.3 |
| Tidak forces and winter monsoon | Neap tide | 16.1 ± 0.2 | 30.7 ± 0.4 | 26.7 ± 0.9 | 23.3 ± 0.7 | 41.6 ± 0.5 | 16.6 ± 0.6 | 7.5 ± 0.6 | 20.2 ± 0.6 |
| | Spring tide | 14.5 ± 1.5 | 30.6 ± 3.8 | 24.9 ± 1.9 | 21.6 ± 1.6 | 39.0 ± 1.8 | 14.7 ± 1.6 | 6.2 ± 1.2 | 18.4 ± 1.6 |
| | average | 15.3 ± 1.6 | 30.7 ± 4.2 | 25.8 ± 2.7 | 22.4 ± 2.2 | 40.4 ± 2.3 | 15.6 ± 2.2 | 6.9 ± 1.8 | 19.3 ± 2.2 |

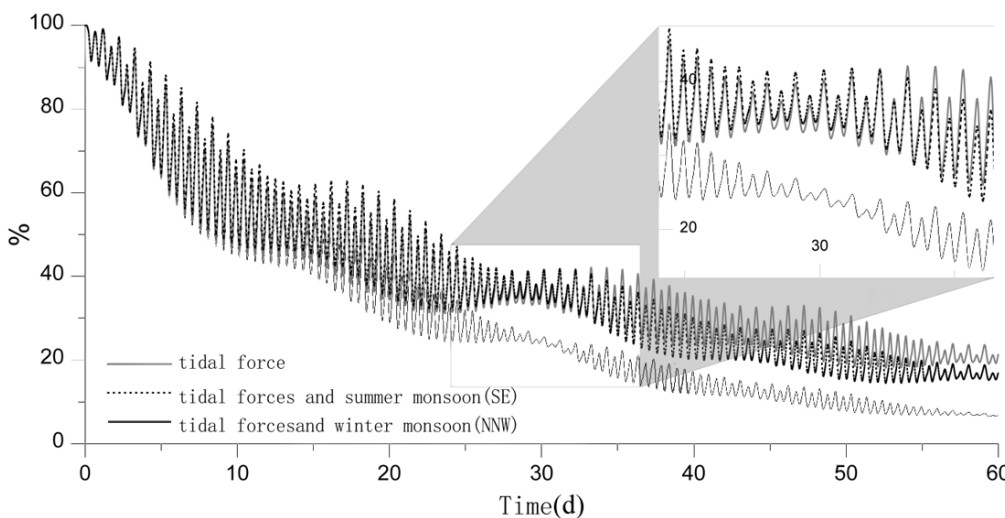

**Figure 4.** The simulated water-exchange process in terms of the percentage of remnant dye.

Residual currents play a leading role in the transport and diffusion of pollutants in the bay. By examining changes in the monthly average structure of the Eulerian residual flow, it was possible to perform a detailed analysis of the distribution of the average residence times in both the entire bay and specific regions under different conditions.

### 3.1. Driven by Tidal Currents Alone (Scenario 1)

Under tidal currents as the only driver, $T_r$ generally increases from the mouth to the head of the bay, with values being greater in the eastern region than in the western region (Figure 5a). The shortest water exchange time occurs at the mouth of the bay (<10 d), indicating a strong water exchange capacity, while the northeastern water bodies of the bay exhibit the longest average residence time (approximately 150 d). This suggests the weakest water exchange capability. The average $T_r$ for the entire bay is 34.4 ± 0.6 d when releasing the tracer during spring tides and 36.5 ± 0.9 d during neap tides, averaging 35.5 ± 1.5 d across both tide periods, which is close to the results of Liu et al. (52 d) [29] and Shi et al. (41 d) [30]. Among them, the value given by Liu et al. [29] is somewhat larger, which is due to the fact that the numerical model (Princeton Ocean Model) does not consider the influence of tidal flats and the deviation caused by different water depths and shorelines used. Regarding the regional distribution of RT in the absence of wind (Figure 5a), Yuan et al. [35], using the FVCOM model, were in agreement with the results of the present study.

The speed of water exchange is primarily controlled by two factors: the distance of the water body from the bay mouth and the structure of the tide-induced residual current field. As the bay mouth is the sole channel connecting Jiaozhou Bay with the outer sea, water masses near the mouth will be preferentially diluted by seawater, resulting in a shorter water exchange time trend in the south and a longer water exchange time trend in the north. As shown in Figure 5b, the water body between the inner and outer Jiaozhou Bay mouths is influenced by three strong residual current systems that easily facilitate exchange with the outer seawater, resulting in an average residence time of only 6.1 ± 0.7 d for the central region of outer Jiaozhou Bay (VII) (Table 1). The water exchange time varies considerably in the sea area north of Huangdao due to a counterclockwise residual circulation that hampers the outward movement of materials in the water. Most materials in the water bodies of the northwest inner Jiaozhou Bay are carried along the eastern shore of Jiaozhou Bay to the bay mouth by a large-scale clockwise residual circulation west of Cangkou Channel. The water bodies of the northeastern region of inner Jiaozhou Bay are primarily controlled by a counterclockwise residual circulation north of Cangkou Channel, making it challenging to transport materials to the open sea, hence the longer average exchange time in the east compared to the west. As illustrated in Figure 5a and Table 1, the average residence time

for the water bodies of the northwestern region of inner Jiaozhou Bay (III) is 52.6 ± 1.7 d while that for the northeastern region of inner Jiaozhou Bay (V) is as high as 91.8 ± 3.4 d.

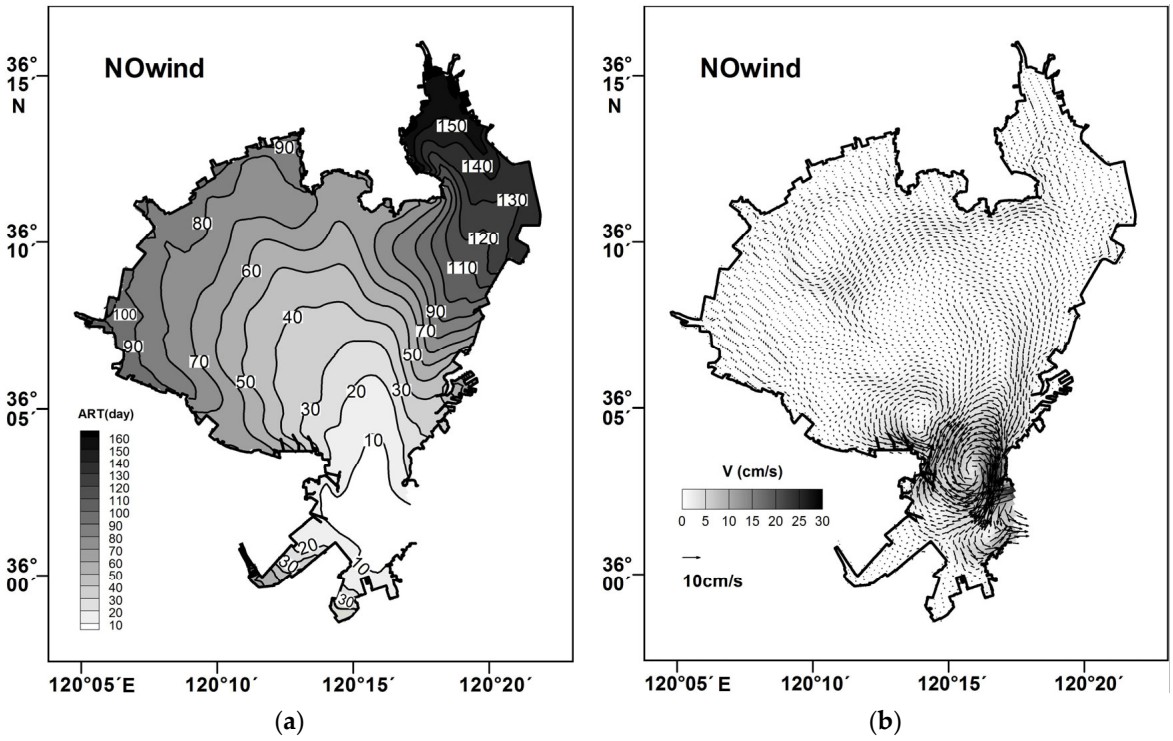

**(a)**                                                 **(b)**

**Figure 5.** The average residence time (ART) distribution and residual current were obtained with tidal forcing only. (**a**) The average residence time; (**b**) The residual current.

### 3.2. Driven Jointly by Summer Monsoon and Tidal Currents (Scenario 2)

Compared to the case with only tidal currents as the driver, the inclusion of the southeast summer monsoon causes significant changes in the $T_r$ distribution (Figure 6a). Its distribution is similar to the distribution of water quality in Jiaozhou Bay in August given in the report on the marine environmental quality of Qingdao in recent years which proves the rationality of the simulation results from the side. Although the increasing trend from the bay mouth to the head remains, the eastern part of the bay now has lower values than the western part. Overall, the water exchange time for the entire bay decreases and the water exchange capability is enhanced, indicating that wind stress plays a significant role in the water exchange process in Jiaozhou Bay.

A further comparison between Figures 5b and 6b reveals that the large clockwise residual circulation west of the Cangkou Channel weakens under the persistent influence of the summer monsoon. Additionally, a strong northward and westward residual current emerges on the western shore of inner Jiaozhou Bay and the southern portion of Hongdao, respectively. As a result, some of the dye is transported towards the head of the bay first, causing a longer stay in inner Jiaozhou Bay. This leads to an extension of the contour lines of the water exchange time towards the head of the bay (Figure 6a). However, the range and intensity of all other residual circulations are enhanced to varying degrees, which will increase the exchange rate of the bay water with the outer sea. This is corroborated by the curve in Figure 4 where the water exchange process starts out slower under the joint influence of the summer monsoon and tidal currents than under the sole influence of tidal currents but it accelerates after one month. The average $T_r$ for the entire bay is 27.9 ± 1.2 d when releasing the tracer during spring tides and 29.9 ± 0.1 d during neap tides, averaging 28.9 ± 1.3 d across both tide periods.

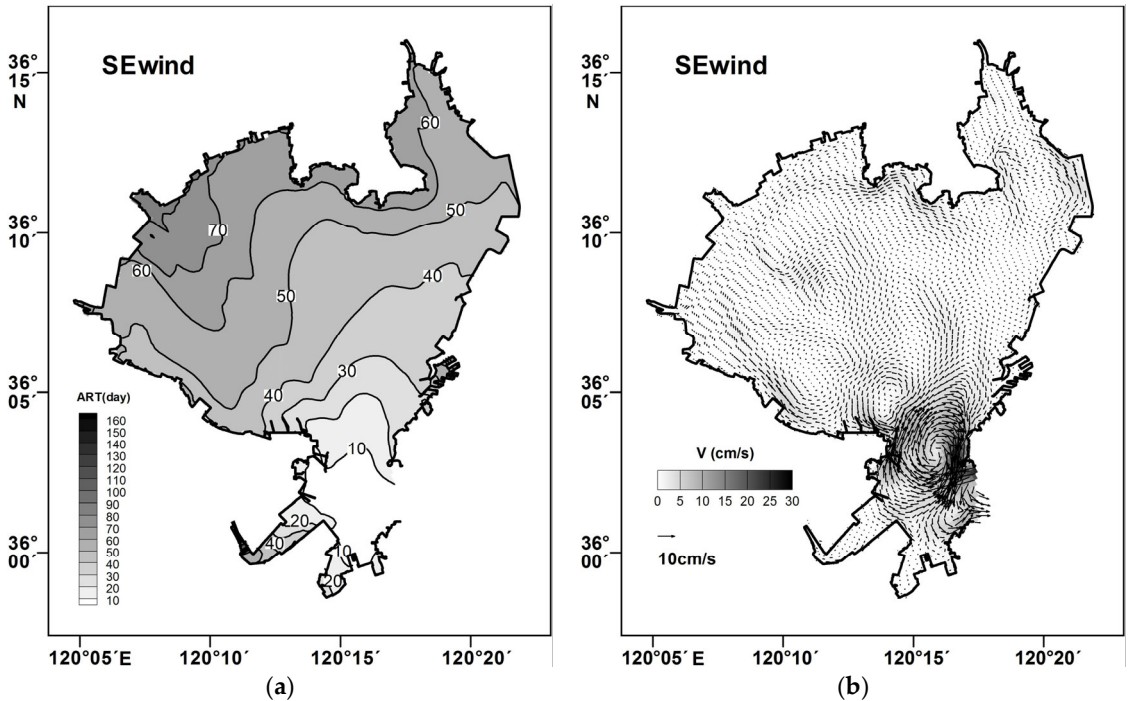

**Figure 6.** The ART distribution and residual current obtained in tidal forcing with summer monsoon. (**a**) The average residence time; (**b**) The residual current.

Due to the changes in the hydrodynamic structure, all areas, except for Qianwan Bay (II), the central region of inner Jiaozhou Bay (VI), and the central region of outer Jiaozhou Bay (VII) where water exchange time increases, experienced a reduction in the water exchange time to varying degrees. In particular, in the northeastern region of inner Jiaozhou Bay (V), $T_r$ declines by more than 50 d (Table 1). Qianwan Bay, with its mouth facing northeast, is adversely affected by southeast winds for water circulation, leading to a reduction in the water exchange capability ($24.7 \pm 1.6$ d). The increased water exchange time in the central region of inner Jiaozhou Bay (VI) is due not only to the weakening of the large-scale clockwise residual circulation west of Cangkou Channel but also due to the influence of the adjacent northeastern region of inner Jiaozhou Bay (V). Under the influence of the southeast wind, more tracers from the northeastern region of inner Jiaozhou Bay (V) enter the central region of inner Jiaozhou Bay, resulting in a longer $T_r$ on average ($28.8 \pm 1.4$ d). In contrast, the $T_r$ for the northeastern region of inner Jiaozhou Bay (V) is reduced to $40.2 \pm 1.5$ d on average. Similarly, due to the influence of adjacent water bodies, the $T_r$ for the central region of outer Jiaozhou Bay (VII) also increases ($6.5 \pm 1.6$ d).

### 3.3. Driven Jointly by Winter Monsoon and Tidal Currents (Scenario 3)

Compared to the summer monsoon, the winter monsoon is more potent. Under the combined influence of NNW winds and tidal currents, the distribution of the water exchange time undergoes drastic changes, presenting a stark contrast to the previous two scenarios (Figure 7b). Similarly, its distribution is close to the distribution of water quality in Jiaozhou Bay in March given in the report on the marine environmental quality of Qingdao in recent years, which proves the rationality of the simulation results to some extent. Generally, the central and western parts of Jiaozhou Bay have a shorter average residence time with a uniform spatial distribution (approximately 20–30 d). The northeastern part has a longer water exchange time than the central and western parts, displaying a steep gradient (20–60 d). The average residence time across the entire bay reduces considerably, indicating a significant enhancement in the water exchange capability (Table 1). The average $T_r$ for the entire bay is only $18.4 \pm 1.6$ d when releasing the tracer during spring tides and $20.2 \pm 0.6$ d during neap tides, averaging $19.3 \pm 2.2$ d across both tide periods.

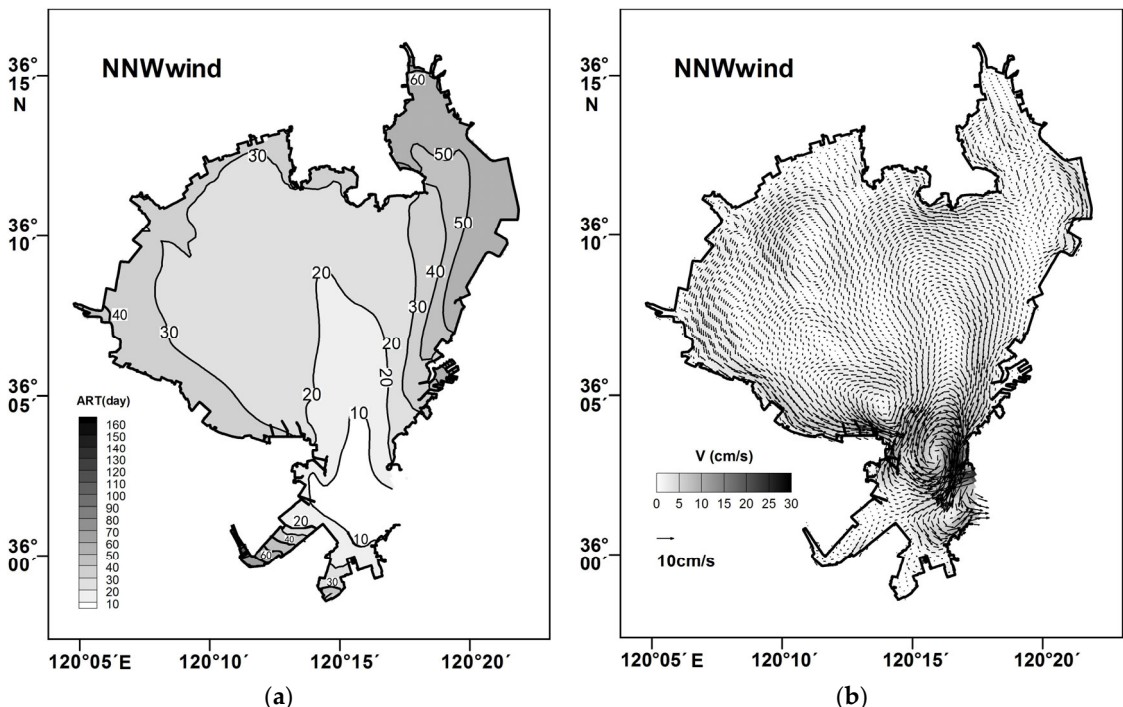

**Figure 7.** The ART distribution and residual current obtained in tidal forcing with winter monsoon. (**a**) The average residence time; (**b**) The residual current.

As depicted in Figure 7b, under the persistent influence of a strong winter monsoon, a powerful southward residual current emerges in the bay, particularly in the coastal areas, that is highly conducive for material transport towards the bay mouth. Except for the southwestern part of the clockwise residual circulation near the mouth of inner Jiaozhou Bay (adjacent to the mouth of Qianwan Bay), which weakens to a slight extent, all residual circulations display a significant increase in range and intensity, particularly the large-scale clockwise residual circulation west of Cangkou Channel. This circulation and the counterclockwise circulation north of Huangdao nearly entirely control the transport of materials in inner Jiaozhou Bay. Additionally, the counterclockwise residual circulation north of Cangkou Channel transitions to a clockwise direction that promotes water exchange. As shown above, the winter monsoon and tidal currents collectively govern the circulations of currents in Jiaozhou Bay and the transport of materials.

Compared to scenarios 1 and 2, the water exchange capability of inner Jiaozhou Bay in scenario 3 greatly enhances, particularly in the northwestern region of inner Jiaozhou Bay (III), the region south of Hongdao (IV), as well as the central region of inner Jiaozhou Bay (VI). In addition, the water exchange capability of the northeastern region of inner Jiaozhou Bay (V) is significantly higher (average $T_r$ of $40.4 \pm 2.3$ d) under scenario 3 than under scenario 1 (average $T_r$ of $91.8 \pm 3.4$ d) but there is no dramatic change between the summer monsoon results of scenarios 2 and 3 (average $T_r$ of $40.4 \pm 2.3$ d vs. $40.2 \pm 1.5$ d). The water exchange capability of outer Jiaozhou Bay weakens to some extent, especially in Haixi Bay (I) ($15.3 \pm 1.6$ d) and Qianwan Bay (II) ($30.7 \pm 4.2$ d). This weakening is attributed to two factors: (1) the weakening of the southwestern part of the residual circulation near the mouth of inner Jiaozhou Bay and (2) the orientation of the two bay mouths that make the northward winds unfavorable for the exchange of water masses in the two bays.

A comparison of the three processes revealed that the spatial distribution of the water exchange time is influenced by the relative distance of the water body from the bay mouth that exhibited an increasing trend from the bay mouth to the bay head. The differences in distribution are primarily caused by changes in the structure and intensity of the residual current field. Although Jiaozhou Bay is predominantly influenced by tidal dynamics, the role of wind stress in the water exchange process should not be overlooked. The water

exchange process in the bay is collectively controlled by the monsoon and astronomical tides. Overall, the bay's water exchange capability is strongest in winter when the bay is influenced jointly by winter monsoon and tidal currents. It is then strongest in summer under the joint influence of the summer monsoon and tidal currents while it is weakest when under the sole influence of tidal currents.

*3.4. Water Exchange in Severely Polluted Areas and Its Impact*

The northeastern coast of Jiaozhou Bay is dotted with numerous rivers and pollution outlets (Figure 1), resulting in significant pollution [6,26]. As shown in Table 1, compared to other areas, the northeastern region (V) has a longer average residence time (>40 d), indicating slower water exchange in this maritime area that exacerbates environmental deterioration. Under the joint action of monsoons and tidal currents, pollutants in the northeastern waters gradually migrate out of the bay, potentially impacting the environment of other maritime areas during this exchange process.

Given the current status of the water exchange capability in this heavily polluted area under different conditions and its impact on other areas, the aforementioned water exchange model was used for simulation. Specifically, the tracer was placed in the northeastern part of the bay, as indicated by the white number 1 in Figure 2, with its concentration in other maritime areas set to 0. The simulation scheme was the same as that for the entire bay. The results of the tracer release during the spring and neap tide periods showed that the average residence time in the northeastern part of Jiaozhou Bay under scenarios 1–3 were $53.6 \pm 6.5$ d (tidal currents), $16.1 \pm 3.0$ d (summer monsoon + tidal currents), and $16.1 \pm 4.5$ d (winter monsoon + tidal currents), respectively. This indicates that under the sole effect of tidal currents, the average residence time is the longest and the water exchange capability is the poorest. Both the southeast summer monsoon and the north–northwest winter monsoon are highly beneficial for the exchange of waters in the northeastern portion. Although the water exchange time did not differ significantly between the two monsoon conditions, the material exchange process did.

To evaluate the impact on other areas, we introduced the "exchange rate", which is defined as the ratio $TC_i(t)$ to $TC_0$, where $TC_i(t)$ represents the total amount of tracer in a particular region, I, at a specific moment, t, in the exchange process, and $TC_0$ signifies the total amount of tracer initially released. This parameter can represent the impact of foreign substances on a region of interest. The order of the peak exchange rates occurring in each region can also indicate the path of material transport.

A comparison of the exchange rate among the different regions under the three scenarios (Figure 8) revealed that (1) the rate of pollutant transport from the northeastern waters decreased in the order of winter monsoon + tidal currents > summer monsoon + tidal currents > tidal currents and that (2) the degree of impact of pollutants in this area on surrounding areas decreased in the order of summer monsoon + tidal currents > tidal currents > winter monsoon + tidal currents.

Driven solely by tidal currents, the pollutant transport in the northeastern waters is primarily controlled by the tide-induced residual currents. Under the influence of the large-scale clockwise residual circulation west of Cangkou Channel, most pollutants are transported along the east coast of the inner Jiaozhou Bay to the bay mouth. During this process, the central region of inner Jiaozhou Bay (VI), the region south of Hongdao (IV), and the northwestern region of inner Jiaozhou Bay (III) are relatively heavily impacted. Subsequently, under the influence of three strong residual circulations, pollutants are gradually exchanged into the open sea. However, the waters along the route are impacted to a lesser extent due to their proximity to the bay mouth and the strong dynamics.

The region south of Hongdao (IV), the northwestern region of inner Jiaozhou Bay (III), and the central region of inner Jiaozhou Bay (VI) are significantly influenced by the combined effects of the southeastward summer monsoon and tidal currents. Due to the weakening of the clockwise residual circulation near Cangkou Channel in terms of both range and intensity as well as the influence of strong westward winds and sea currents

along the coast, a considerable amount of pollutants first enter region IV and then swiftly reach region III. Some pollutants are gradually transported outwards along the eastern coast, impacting the central region of inner Jiaozhou Bay (VI) during the transport process. As shown in Figure 8, the exchange rate peaks first in region IV, followed by region III and then region VI. Furthermore, as indicated in the previous section, both regions IV and III have a long average residence time and a weak water exchange capability. Under the influence of the southeastward summer monsoon, these two regions are impacted by pollutants from the northeastern waters, making it more difficult to achieve environmental management and protection of the aquaculture and tourism areas of Hongdao as well as the Daguhe and Yanghe wetland conservation areas.

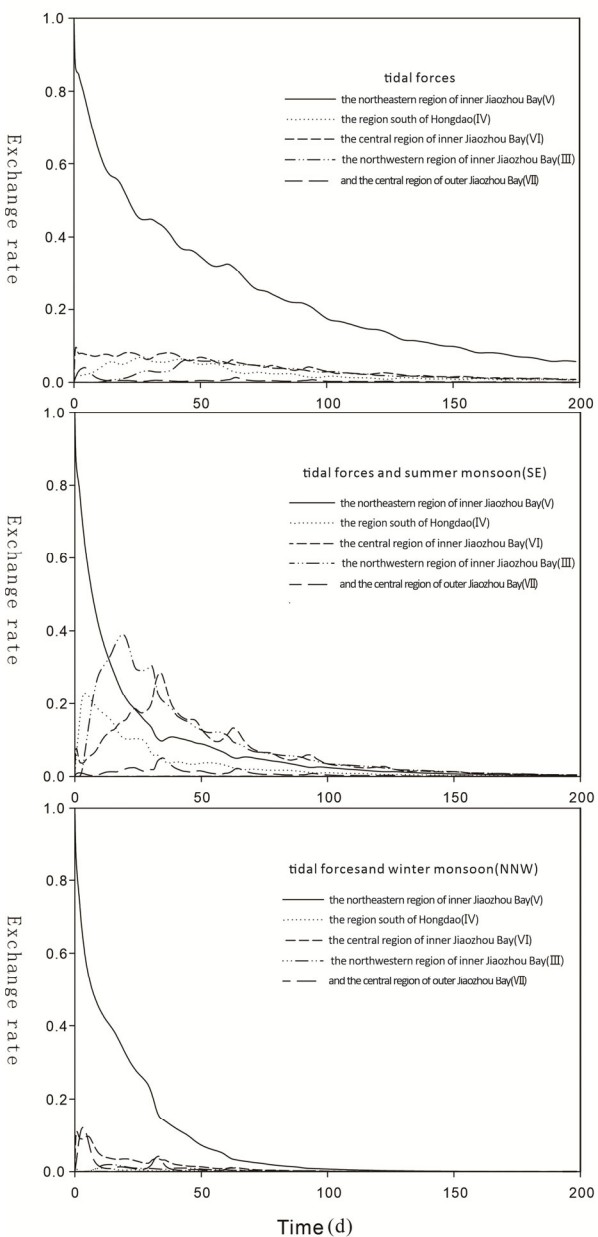

**Figure 8.** The percentage exchange of Jiaozhou Bay, China.

Under the combined action of a strong north–northwestward winter monsoon and tidal currents, the material transport in Jiaozhou Bay is collectively controlled by tide-induced residual currents and wind-driven sea currents. Most pollutants from the northeast are rapidly transported to the bay mouth along the eastern coast of Jiaozhou Bay, affecting

only the central region of inner Jiaozhou Bay (VI) and the central region of outer Jiaozhou Bay (VII). However, due to the strong water exchange capability of these two regions, the pollutants are quickly exchanged out to the open sea, causing minimal impact on other waters.

In summary, when the summer monsoon is present, the exchange rate of pollutants in the northeastern waters is fast, having the greatest impact on other areas. Under the sole influence of tidal currents, the exchange is relatively slow, causing a less but more prolonged impact on surrounding areas. The fastest water exchange and minimal impact occur when a winter monsoon is present.

*3.5. The Implications and Limitations of the Study*

Semi-enclosed bays not only have wide waters, weak winds, and small waves but also have long and narrow tidal inlets with large water depths and slow siltation of tidal flats. They are ideal places for human survival and development. For example, Tokyo Bay in Japan and San Francisco Bay in the United States have formed two famous bay areas in the world. Therefore, the comprehensive development and utilization of bay resources play a particularly important role in the development and utilization of marine resources. However, because it is located in the transition area between the mainland and the ocean, the bay integrates the material energy and physical and chemical characteristics of the two systems of land and sea and the ecological environment has potential instability, vulnerability, and variability. At the same time, due to the motion characteristics of the water body, various factors are highly correlated and interact with each other, which can easily lead to environmental degradation and resource damage. Jiaozhou Bay is a typical semi-closed shallow bay affected by the monsoon climate. The findings of this study not only provide scientific support for the rational development, scientific management, and comprehensive utilization of Jiaozhou Bay but also provide reference for other similar bays in the world. For example, the reasonable season and time of urban sewage discharge into the sea can be selected according to the influence of the monsoon climate on the water exchange capacity of the bay.

However, similar to most previous studies on bay water exchange [27–33], this paper is also based on numerical models and lacks the support of field-measured data. The wind stress used in the model is driven by the seasonal average wind field. In recent years, with global warming, extreme weather events such as strong winds and heavy rains have occurred frequently. In addition, the intensification of human development activities such as reclamation, reservoir, and urban impervious surface expansion around the bay has caused the concentration of precipitation in the basin to inject into the bay for a short time. It is very necessary to study the process and mechanism of bay water exchange driven by climate change and human activities. In addition to tidal current and wind, more factors such as seasonal short-term rainfall and surface runoff should be considered. This may require the deployment of artificial tracer buoys with positioning and marine element monitoring functions on site combined with the Lagrangian particle tracking numerical model to explore the three-dimensional water exchange process in the bay driven by multiple factors.

## 4. Conclusions

Based on the definition of average residence time and considering the influence of tracer release timing, this study simulated the distribution of water exchange times in Jiaozhou Bay under three scenarios. Based on bay zoning, the effects of monsoons and tidal currents on global and local water exchange processes were examined in detail. The primary conclusions are as follows:

(1) The water exchange process in Jiaozhou Bay is influenced by both astronomical and meteorological factors, with the water exchange capability decreasing in the order of winter monsoon + tidal currents > summer monsoon + tidal currents > tidal currents.

Changes in the structure and intensity of the residual current field were the primary cause for the exchange rate differences;

(2) The moment of tracer release has little effect on the average water residence time. The spatial distribution of the water exchange time is affected by the distance of the water body relative to the mouth of the bay, with the water exchange time increasing from the mouth to the head of the bay. Under the sole influence of tidal currents, the water residence time averaged across both spring and neap tides is 35.5 $\pm$ 1.5 d for the entire bay and the average is greater for the eastern bay waters than for the western bay waters. Under the joint influence of a summer monsoon and tidal currents, the following observations were made: (a) the water exchange process of the entire bay starts slowly and then accelerates; (b) the water exchange time increases from the mouth to the head of the bay and it is shorter in the eastern part than in the western part, especially the northeastern region of inner Jiaozhou Bay with the largest decrease; and (c) the average water residence time of the entire bay is 28.9 $\pm$ 1.3 d. In contrast, the joint influence of a winter monsoon and tidal currents leads to a significant increase in the water exchange capability of the entire bay and a sharp variation in the spatial distribution of the water exchange time. (a) The water exchange time is relatively short and uniformly distributed in the central and western parts; (b) it is generally higher in the eastern part than in the central and western parts while it displays a large gradient in the eastern part; and (c) it averages 19.3 $\pm$ 2.2 d across both spring and neap tides for the entire bay;

(3) In the heavily polluted northeastern region, water exchange under the sole influence of tidal currents is relatively slow, with an average water exchange time of 53.6 $\pm$ 6.5 d and a small yet relatively long-lasting impact on the surrounding areas. Water exchange is faster under the joint influence of a summer monsoon and tidal currents, with an average water exchange time of 16.1 $\pm$ 3.0 d and the greatest impact on other areas. In addition, water exchange is fastest under the joint influence of a winter monsoon and tidal currents, with an average water exchange time of 16.1 $\pm$ 4.5 d and the least impact on the surrounding areas.

**Author Contributions:** Conceptualization, Z.T., J.S. and W.W.; Methodology, W.W. and F.L.; validation, J.S., Y.L. and Y.S.; Data curation, J.S. and C.L.; Writing—original draft preparation, Z.T., W.W. and J.S.; Writing—review and editing, Z.T., W.W. and F.L. All authors have read and agreed to the published version of the manuscript.

**Funding:** This research was supported by the National Natural Science Foundation of China (No. 62171435) and the Open Fund of the No. 1 Institute of Geology and Mineral Resources of Shandong Province (No. 2020DW02).

**Institutional Review Board Statement:** The study did not require ethical approval.

**Informed Consent Statement:** Not applicable.

**Data Availability Statement:** The numerical data used to support the findings of this study are included within the article.

**Conflicts of Interest:** The authors declare no conflict of interest.

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
