# Peer review of "Response of Sea Water Exchange Processes to Monsoons in Jiaozhou Bay, China"

_sustainability, doi:10.3390/su152115198_

Round 1

Reviewer 1 Report

This manuscript was well written, it can be accept for publication. I have one minor suggestion for the authors' reference, but it's not a big problem. The authors discuss the influence of winter and summer monsoon, it's mainly about the monsoon wind but not precipitation. I think it can be further clarified in the manuscript. One the other hand, i have a question, is the effect of monsoon precipitation also evaluated in the model?

Author Response

Dear reviewer,

Thank you for your comments. We accept your suggestions and conduct further research and make revisions. According the comments, we have carefully revised the manuscript and all the revisions were marked in red font and highlighted in yellow in the revised manuscript, and some academic issues have also been explained.Please see the attachment.

Reviewer 2 Report

Dear Authors,

Thank you very much for the review invitation of the manuscript.

I read your paper " Response of water exchange processes to monsoons in Jiaozhou 2 Bay, China" with great interest and well-written manuscript

To make the manuscript more complete, I only need to make minor changes.

Please offer more publications pertaining to your subject fields because your literature review was inadequate.

2nd Figure: Please include a wind direction indicator, a scale bar, and a legend.

I suggest putting "sea" before "water" in the title, for example, "Response of sea water exchange processes to monsoons in Jiaozhou Bay, China."

The author must indicate that the monsoon that the author is concerned with is wind and tide, and whether or not rain and rainfall-runoff elements are addressed.

Author Response

(The authors gave the same response as above.)

Reviewer 3 Report

Main notes:

1. In the manuscript, in the Discussion section, the authors do not discuss the results obtained in the context of international experience by comparing these results with the results of other similar studies in other regions of China and the world. In this regard, the Discussion section looks poor. How might your results be of interest to the international scientific community?

2. Modeling is just one of the approaches to obtaining new knowledge. What about comparing your results with those of other studies that used independent methods? Why should the reader trust your results? The authors should be more critical in their conclusions.

3. I did not find a paragraph in the manuscript discussing the limitations of the study.

Small comments:

(a) I recommend not placing the bar chart in Table 1, but making it a separate figure.

(b) Table 1. At what probability were the confidence intervals calculated? 95%? Write about it.

The English language of the manuscripts needs improvement.

Author Response

(The authors gave the same response as above.)

Reviewer 4 Report

Minor editing of English language required.

Author Response

(The authors gave the same response as above.)

Round 2

Reviewer 3 Report

Dear Authors,

Thank you for your answers and revision of the manuscript.

Some improvement in English is required.

Reviewer 4 Report

The Authors have addressed my concerns carefully. This revised manuscript can be accepted in present form. Refinements in terms of style and English language can be made at the proofreading stage.

Minor editing of English language required.